# Residual Context Diffusion Language Models

Yuezhou Hu [* 1]   Harman Singh [* 1]   Monishwaran Maheswaran [* 1]
Haocheng Xi [1]   Coleman Hooper [1]   Jintao Zhang [1]   Aditya Tomar [1]
Michael W. Mahoney [1]   Sewon Min [1]   Mehrdad Farajtabar [2]
Kurt Keutzer [1]   Amir Gholami [† 1]   Chenfeng Xu [† 1]

**Links:** Code | Models | Project Page

## Abstract

Diffusion Large Language Models (dLLMs) have emerged as a promising alternative to purely autoregressive language models because they can decode multiple tokens in parallel. However, state-of-the-art block-wise dLLMs rely on a "remasking" mechanism that decodes only the most confident tokens and discards the rest, effectively wasting computation. We demonstrate that recycling computation from the discarded tokens is beneficial, as these tokens retain contextual information useful for subsequent decoding iterations. In light of this, we propose Residual Context Diffusion (RCD), a module that converts these discarded token representations into contextual residuals and injects them back for the next denoising step. RCD uses a decoupled two-stage training pipeline to bypass the memory bottlenecks associated with backpropagation. We validate our method on both long CoT reasoning (SDAR) and short CoT instruction following (LLaDA) models. We demonstrate that a standard dLLM can be efficiently converted to the RCD paradigm with merely ∼300 million tokens. RCD consistently improves frontier dLLMs by 4–11 percentage points in accuracy with minimal extra computation overhead across a wide range of benchmarks. Notably, on the most challenging AIME tasks, RCD nearly doubles baseline accuracy and attains up to 4–5x fewer denoising steps at baseline's peak accuracy.

---

[*]Equal contribution [†]Equal Advising [1]University of California, Berkeley [2]Apple. Correspondence to: Chenfeng Xu <xuchenfeng@berkeley.edu>, Amir Gholami <amirgh@berkeley.edu>.

*Proceedings of the 43rd International Conference on Machine Learning*, Seoul, South Korea. PMLR 306, 2026. Copyright 2026 by the author(s).

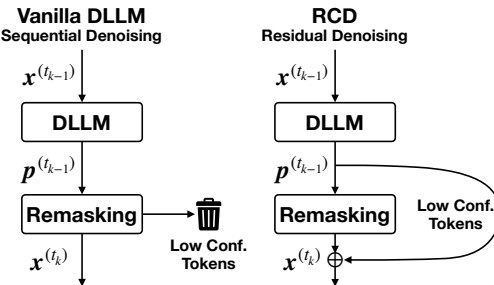

*Figure 1.* Overview of the residual denoising mechanism in Residual Context Diffusion (RCD). Remasking happens during each denoising step, discarding low-confidence token even though they are already explicitly computed. To recycle computation from previous steps, our proposed method extracts information before it is discarded, and forwards it to the next denoising step.

## 1. Introduction

Diffusion Large Language Models (dLLMs) have recently emerged as a promising alternative to purely autoregressive (AR) models, showing encouraging results on instruction following (Nie et al., 2025; Ye et al., 2025), code generation (Gong et al., 2025), long-context understanding (Liu et al., 2025), and complex reasoning (Wang et al., 2025d; Zhao et al., 2025). While AR models currently dominate large-scale industrial deployment, what makes dLLMs particularly attractive today is their parallel decoding capability: instead of generating tokens strictly one-by-one like AR, they update multiple tokens simultaneously, offering a path to move decoding away from the memory-bandwidth-limited regime that often dominates AR inference (Williams et al., 2009), toward higher compute utilization (Kim et al., 2025). Indeed, recent systems have reported notable speedups over strong AR baselines (Cheng et al., 2025; Wang et al., 2025c).

However, today's dLLMs still trail AR models in accuracy. This gap is unsatisfactory: AR decoding generates each token in a single pass, whereas dLLMs typically expend significantly more computation per token via multiple sequential denoising iterations. We observe that in state-

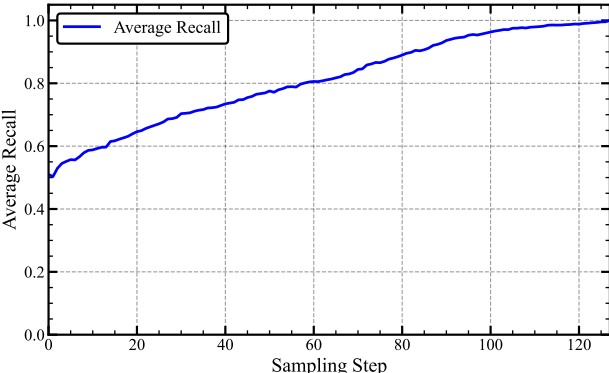

*Figure 2.* Average token recall across LLaDA GSM8K sampling steps. For each step $k$ (x-axis), recall@5 (y-axis) is the fraction of final decoded tokens that appear in the step-$k$ top-5 predictions at the same position. High early-step recall@5 indicates that intermediate distributions contain semantically informative signals.

of-the-art block-wise dLLMs, this extra compute does not reliably translate into accuracy gains. While this gap is partly rooted in the inherent challenges of training diffusion models to capture the strict left-to-right causal dependencies, it is further exacerbated by the inference-time "remasking" strategy, which commits only the most confident tokens in each iteration and discards the rest, as shown in Fig. 1(left). This effectively "wastes" the intermediate computation performed on tokens that remain undecoded. In Fig. 2, we show that these intermediate distributions can indeed provide crucial guidance toward the final denoised sequence. In other words, much of the additional compute is wasted, even though it contains structured signals regarding global context. This motivates us to raise a fundamental question:

> *How can we fully exploit the information from every denoising step to unleash the potential of dLLMs?*

Inspired by the residual learning paradigm (He et al., 2015), we propose **Residual Context Diffusion (RCD)**, a dLLM with a residual denoising mechanism (Fig. 1(right)) that leverages low-confidence tokens instead of discarding them. By treating their latent representations as a residual update to the input context, the model propagates not only discrete tokens but also continuous embedding vectors enriched with contextual information (Eq. (2)). This allows the model to progressively refine and crystallize its knowledge, transforming previously wasted computation into a guiding signal.

Although the idea of reusing intermediate computation in block-wise diffusion decoding appears seemingly simple, there are several reasons why making it work is far from trivial. *(1) It is unclear where residual contextual signals should come from and how they should be aggregated.* One could feed back the token prediction itself, but richer infor-

mation lives in the full predictive distribution since decoding collapses a distribution into a one-hot decision, and even more semantic cues may be encoded in the model's hidden states. More importantly, naively injecting and aggregating such residual context can distort the denoising input distribution, leading to instability or degraded decoding. *(2) Training with residual feedback is intrinsically hard.* The self-referential loop creates a long unrolled computation graph reminiscent of RNNs, making backpropagation-through-time prohibitively expensive under realistic memory budgets.

To address these issues, we propose Residual Context Diffusion, which equips diffusion LLMs with the mechanism for recycling computation via residual information.

- **Entropy-Based Embedding Aggregation:** For context selection and aggregation, prior approaches often reuse hidden states (e.g., AR-style such as Coconut (Hao et al., 2025) and diffusion-style such as Loopholing (Jo et al., 2025)). However, this is suboptimal: it fails to account for masked-vs-unmasked structure, and it can suffer from a mismatch between embedding magnitudes. RCD instead constructs residual context based on the model's own embedding codebooks, and it introduces entropy-based aggregation, *i.e.,* using normalized Shannon entropy with temperature-adjusted alignment to extract reliable signals from residual distributions and bridging train–test mismatch.

- **A two-stage training pipeline:** We use a memory- and data-efficient two-stage training pipeline to train our framework. In the first stage, we train a lightweight reference model that generates residual probabilities, and in the second stage we use the reference model to generate model's input context. The two-stage pipeline decouples the training target and successfully overcomes the memory bottlenecks of backpropagating. We demonstrate that our pipeline can efficiently convert a standard dLLM into the RCD paradigm using as few as ∼300 million tokens.

- **A new Pareto knob:** Residual context allows for a new trade-off between denoising steps and residual transmission. RCD achieves consistently better accuracy–latency trade-offs than throughput-matched baselines, with minimal extra computation compared to the dLLM backbone. Evaluated on the SDAR family and LLaDA, RCD yields 4–11 percentage gains on benchmarks like GSM8K and MATH500. Notably, on AIME24/25, RCD nearly doubles baseline accuracy while requiring up to 4–5× fewer steps at equivalent accuracy levels.

## 2. Preliminaries

This section establishes the preliminaries for our method. We first introduce dLLMs and their masked denoising mechanism; and we then review the soft token method in la-

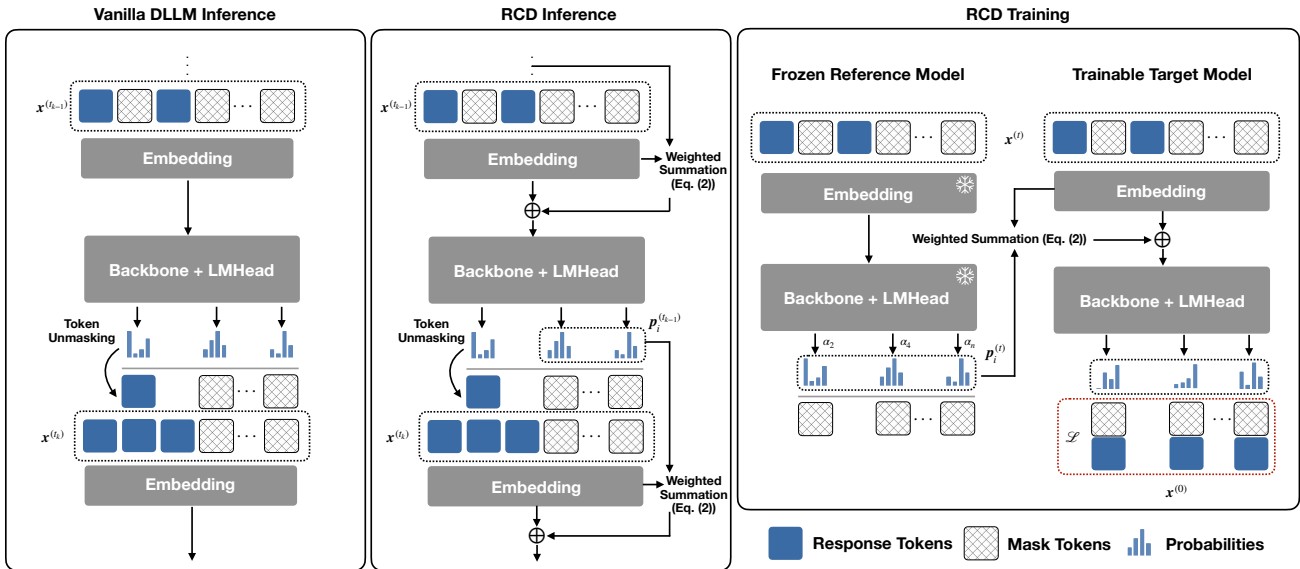

*Figure 3.* **(Left)** In vanilla dLLM inference, tokens not selected are reverted to static mask embeddings for the next step, discarding the predicted probability information. **(Center)** RCD Inference preserves this information by calculating a weighted summation of the unselected probability distributions (residuals) and injects them into the input embeddings of the subsequent step. This allows the model to aggregate more information and refine ambiguous tokens over time. **(Right)** RCD Training employs a decoupled strategy. A Frozen Reference Model generates stable probability targets and entropy weights ($\alpha$). These are used to construct the residual vectors, which are then added to the Trainable Target Model's embeddings, helping the model to effectively leverage residual context for prediction.

tent reasoning studies that is highly related to our proposed model. A full list of related works is in Appendix A. Note that we use superscript $(t)$ to denote the denoising time step and the subscript $i$ to represent the token position.

## 2.1. Diffusion Large Language Models

Diffusion-based LLMs treat text generation as a progressive denoising process within a masked latent space. They initialize all tokens with identical masks [M], and in each forward pass the model selects the highest-confidence token for decoding, culminating in a fully denoised output.

**Training.** Without loss of generality, assume the input $\boldsymbol{x}^{(0)} = [x_1, x_2, \dots, x_b]$ is a single block of $b$ tokens. The dLLM is trained to recover $\boldsymbol{x}^{(0)}$ from a noisy sequence $\boldsymbol{x}^{(t)}$, where

$$x_i^{(t)} = \begin{cases} x_i^{(0)} & \text{if } m_i^{(t)} = 0 \\ [\text{M}] & \text{if } m_i^{(t)} = 1 \end{cases}.$$

Here, $m_i^{(t)} \sim \text{Bernoulli}(t)$ is a binary mask sampled independently for each position $i \in \{1, \dots, b\}$, indicating that the token at position $i$ is replaced by the mask [M]. The training objective is to minimize the expected cross-entropy loss over all masked positions. Given the model parameters $\theta$, the loss function is formulated as:

$$\mathcal{L}(\theta) = \mathbb{E}_{\boldsymbol{x}^{(0)}, t} \left[ \frac{1}{t} \sum_{i: m_i^{(t)} = 1} - \log P_\theta(\hat{x}_i = x_i^{(0)} \mid \boldsymbol{x}^{(t)}) \right].$$

By sampling $t \sim \mathcal{U}(0, 1)$, the model learns to reconstruct tokens across varying levels of corruption.

**Inference.** Inference in a dLLM is a reverse denoising process. Starting from an entirely masked sequence $\boldsymbol{x}^{(1)} = \{[\text{M}]\}^b$, the model iteratively recovers tokens over $K$ iterations. Each follows three primary sub-steps:

- **Prediction:** The model predicts $\boldsymbol{p}_i^{(t_k)}$ for all currently masked positions $i$, where $x_i^{(t_k)} = [\text{M}]$. Here, $\boldsymbol{p}_i^{(t_k)} = [p_{i,1}^{(t_k)}, p_{i,2}^{(t_k)}, \dots, p_{i,V}^{(t_k)}]^\top$ is the $i$-th token's discrete probability distribution and $V$ is the vocabulary size.

- **Selection:** A confidence score $c_i^{(t_k)} = \max(\boldsymbol{p}_i^{(t_k)})$ is calculated for each position $i$. The model selects the top-$m$ positions with the highest confidence to be "committed" or fixed.

- **Update:** For the selected $m$ positions, new tokens $\hat{x}_i$ are sampled from the predicted distribution. The remaining less confident positions are "remasked" by being reset to [M] for the subsequent iteration $t_{k+1}$.

This iterative process continues until the block is fully denoised. However, as noted in Sec. 1, standard dLLMs discard $\boldsymbol{p}_i^{(t_k)}$ for all $i \notin$ top-$m$ at each step, resulting in significant information loss that RCD aims to mitigate.

## 2.2. Soft Tokens

Soft tokens are similar to the model's input embedding vectors that can represent a mixture state of multiple tokens.

Butt et al. (2025); Zhang et al. (2025b) propose to convert discrete probability distributions into soft tokens by taking weighted sum over the vocabulary embeddings: $e = \sum_{j=1}^{V} p_j \boldsymbol{E}_{j,:} = \boldsymbol{E}^{\top} \boldsymbol{p}$, where $\boldsymbol{p} = [p_1, p_2, \ldots, p_V]^{\top}$ is a discrete distribution over the vocabulary, and $\boldsymbol{E} \in \mathbb{R}^{V \times D}$ denotes the embedding codebook. Here, $D$ denotes embedding dimension, and $\boldsymbol{E}_{j,:}$ corresponds to the discrete embedding vector for the $j$-th token in the vocabulary. For RCD, by leveraging this weighted sum, we enable the model to maintain fine-grained contextual information before a token is fully determined. However, naively adding the soft tokens to input embeddings disrupts the discrete masking scheme dLLMs rely on and creates unstable recursive dependencies during training, leaving injecting residual context into dLLMs an open question for RCD to solve.

## 3. The RCD Method

In this section, we introduce Residual Context Diffusion that leverages remasked tokens in dLLMs to enhance generation quality. At the core of RCD is the entropy weighted residual, which dynamically adjusts the weight of residual information based on the probability distribution of each token. We first present the design and rationale behind the entropy weighted residual, followed by a description of the training and inference procedures for RCD. The pipeline of RCD training and inference is shown in Fig. 3. The complete training and inference procedure is summarized in Appendix B.

### 3.1. Entropy Weighted Residual

Here, we formalize the calculation of residual context and define how it is incorporated into the model's input. Following Sec. 2.2, we first obtain the residual information vector by taking a weighted sum of the predicted probability distribution $\boldsymbol{p}_i$ with the input embedding codebook. Specifically, for each token at position $i$ and denoising step $t_k$, we define the generated residual information $\boldsymbol{\Delta}_i \in \mathbb{R}^D$ as

$$\boldsymbol{\Delta}_i^{(t_k)} = \sum_{j=1}^{V} p_{i,j}^{(t_k)} \boldsymbol{E}_{j,:}. \tag{1}$$

Then, we inject $\boldsymbol{\Delta}_i^{(t_k)}$ into the model's input for the next step. Specifically, we only inject the residual information into mask tokens, since these positions need contextual prior to sustain the semantic meaning of the whole sequence. Note that directly applying the standard residual method by plain summation would yield a misaligned magnitude of input hidden states. Thus, we adopt highway connection (Srivastava et al., 2015a;b) as a generalized form of residuals (Team et al., 2026). The highway connection we use interpolates the current masked tokens with the residual information from the previous step $t_{k-1}$ to create $\tilde{e}_i^{(t_k)} \in \mathbb{R}^D$,

the model's input embedding vector:

$$\tilde{e}_i^{(t_k)} =$$
$$\begin{cases} (1 - \alpha_i^{(t_{k-1})}) E(x_i^{(t_k)}) + \alpha_i^{(t_{k-1})} \boldsymbol{\Delta}_i^{(t_{k-1})}, & x_i^{(t_k)} = [\text{M}] \\ E(x_i^{(t_k)}), & x_i^{(t_k)} \neq [\text{M}] \end{cases} \tag{2}$$

where $E(\cdot)$ is the input embedding layer, and $\alpha_i \in [0, 1]$ controls the residual contribution.

Hence, the only question yet to be solved is to determine how much should residual information contribute. In line with recent studies (Wang et al., 2025b), high-entropy tokens play a crucial role in the generation process, due to their ability to carry more structured information compared to low-entropy counterparts. Specifically, in dLLMs, Fu et al. (2025) demonstrate that prioritizing decoding of low-confidence tokens (those with high entropy) can lead to higher confidence in the remaining token predictions and reduce the total number of required decoding steps.

Motivated by this observation, our goal is to dynamically adjust the contribution of discarded (low-confidence) tokens during the residual decoding process. Intuitively, if a token exhibits high entropy, it may contain more critical information that significantly influences subsequent denoising steps. Therefore, we assign a higher weight to such residual information. To implement this mechanism, we calculate each token's normalized Shannon entropy, which ensures $\alpha_i^{(t_k)} \in [0, 1]$:

$$\alpha_i^{(t_k)} = \hat{H}(\hat{x}_i^{(t_k)}) = \frac{H(\hat{x}_i^{(t_k)})}{\log V} = \frac{-\sum_{j=1}^{V} p_{i,j}^{(t_k)} \log p_{i,j}^{(t_k)}}{\log V}. \tag{3}$$

Here, $H(\hat{x}_i^{(t_k)})$ is the standard Shannon entropy of predicted token $\hat{x}_i^{(t_k)}$, and $\log V$ is the maximum possible entropy value, achieved when all tokens are equally likely (uniform distribution over a vocabulary size $V$). This normalized entropy later serves as weight value in Eq. (2). This formulation enables RCD to maintain a richer context throughout the denoising process by incorporating residual information in a principled, entropy-driven manner.

### 3.2. Training

Training RCD presents a unique challenge due to the recursive nature of the residual mechanism. As defined in Eq. (2), the input for the current step depends on the residual output from the previous step. In a naive end-to-end training setup, this creates a circular dependency: the model needs to learn how to generate high-quality residuals and how to use them, jointly. This would require unrolling the denoising steps and performing backpropagation-through-time, which is computationally prohibitive and unstable. To overcome this, we decouple the residual generation from

its utilization. We propose a two-stage training strategy using a lightweight Reference Model to provide stable, proxy residual signals. This approach allows us to train the Target Model via standard single-step supervision without complex graph unrolling.

**Step 1: Reference Model Initialization.** In the first stage, we require a model capable of generating reliable probability distributions to serve as "ground truth" residuals. We define this as the Reference Model ($\mathcal{M}_{\text{ref}}$). We initialize $\mathcal{M}_{\text{ref}}$ from a pre-trained lightweight dLLM and fine-tune it on the downstream dataset using the standard masked objective. Once trained, $\mathcal{M}_{\text{ref}}$ serves as a frozen "hint." Given any masked input $x^{(t)}$, it can estimate a high-quality probability distribution $p^{(t)}$, which represents the ideal contextual information available at noise level $t$.

**Step 2: Residual-Aware Target Training.** In the second stage, we train the Target Model ($\mathcal{M}_{\text{target}}$) to effectively incorporate residual information. This model is also initialized from a base model and is trained with the following recipe. Here, our key idea is to use the frozen $\mathcal{M}_{\text{ref}}$ to simulate the residual signal that would have come from a previous step. During each training step with a masked sequence $x^{(t)}$, the process proceeds as follows:

- **Signal Generation:** The frozen $\mathcal{M}_{\text{ref}}$ computes the probability distributions $p^{(t)}$ and the corresponding entropy weights $\alpha^{(t)}$ (via Eq. (3)).
- **Residual Injection:** We construct the residual vector and input embedding via Eq. (1-2). Crucially, while the probabilities come from $\mathcal{M}_{\text{ref}}$, we perform the weighted sum using the input embedding codebook of $\mathcal{M}_{\text{target}}$. This ensures the residual vector lies in the correct latent space for the Target Model.
- **Optimization:** The Target Model receives the combined input (masked tokens + residual vectors) and is trained to predict the original tokens $x^{(0)}$.

The optimization objective is the standard cross-entropy loss:

$$\mathcal{L} = \mathbb{E}\left[\frac{1}{t}\sum_{i:m_i=1} -\log P_{\theta_{\text{target}}}(x_i^{(0)} \mid \{\tilde{e}_{i'}^{(t)}\}_{i'=1}^{b})\right],$$

where $\tilde{e}_i^{(t)}$ is the input embedding augmented with residual signals from $\mathcal{M}_{\text{ref}}$. By freezing $\mathcal{M}_{\text{ref}}$, we provide a stationary target for $\mathcal{M}_{\text{target}}$, preventing the self-reinforcing instability. This effectively transforms the recursive definition into a supervised learning task for the model to learn to make accurate predictions given a feasible residual context.

### 3.3. Inference

During inference, RCD transitions from the teacher-guided training setup to a self-referential recursive loop. This introduces two key challenges: initializing the residual

stream effectively; and bridging the distribution gap between the proxy signals seen during training (from $\mathcal{M}_{\text{ref}}$) and the model's own predictions. We address these through a "Warm Start" strategy and Temperature-Scaled Entropy.

**Temperature-Scaled Entropy Alignment.** In the training phase, the residual signals $\Delta^{(t)}$ are derived from the fixed, high-quality Reference Model. However, during inference, the Target Model $\mathcal{M}_{\text{target}}$ must generate its own residuals based on its predictions from the previous step $t_{k-1}$. Since $\mathcal{M}_{\text{target}}$ may produce probability distributions that are sharper or flatter than the Reference Model, the resulting entropy weights $\alpha$ can drift, leading to suboptimal residual injection. To align the inference-time statistics with the training distribution, we introduce a residual temperature scalar $T_{\text{res}}$. We adjust the "softness" of the probability distribution used for residual calculation:

$$p_i^{(t_k)}(T_{\text{res}}) = \text{softmax}(z_i^{(t_k)}/T_{\text{res}}),$$

where $z_i$ are the output logits. The entropy weight is then computed on this scaled distribution: $\alpha_i^{(t_k)} = \hat{H}(p_i^{(t_k)}(T_{\text{res}}))$. By tuning $T_{\text{res}}$, this acts as a calibration mechanism. If the model is over-confident (producing negligible entropy), increasing $T_{\text{res}}$ forces the model to attend to the residual context more heavily, matching the behavior learned during the proxy training phase.

**Inference Pipeline.** The overall inference procedure operates as a hand-off process, structured as follows:

- **Initialization:** At the very first denoising step ($t_1$), no previous residual exists. To jump-start the process, we provide two different initialization methods for residual context. (1) Warm start: We "warm up" the process by invoking $\mathcal{M}_{\text{ref}}$ once to generate the initial probability distribution $p^{(t_0)}$. (2) Cold start: We initialize the residual context to zero at the very first denoising step following the common practice in recurrent architectures (Wenke & Fleming, 2019).
- **Recursive Decoding:** From step $t_2$ onwards, $\mathcal{M}_{\text{target}}$ enters a self-loop. Specifically, the model consumes input embeddings augmented by the residual context from the previous iteration (Eq. (2)). Upon generating the current step's logits, we apply the temperature scalar $T_{\text{res}}$ to derive the entropy weight $\alpha^{(t_k)}$, which are subsequently injected into the input for the next denoising step $t_{k+1}$.

## 4. Experiments

### 4.1. Experimental Setup

Here, we describe our experimental setup; further details may be found in Appendix C.

**Models and Datasets.** We evaluate RCD across two distinct dLLM paradigms: LLaDA (Nie et al., 2025), a bidirec-

*Table 1.* Accuracy comparison of SDAR models across mathematical reasoning benchmarks. Results are reported at a confidence threshold of 0.85, grouped by model size (4B, 8B) and block size ($b = 32, 64$ for KV cache-reuse). The Chat variant serves as the initialization for both standard Sequential Denoising (SeqD) and RCD reasoning models. The performance gap between the first and subsequent rows reflects the efficacy of our reasoning-focused adaptation. Compared to the SeqD baseline, RCD consistently yields superior performance, with the most pronounced gains observed in competition-level benchmarks (AIME24/25), where accuracy often more than doubles. Evaluation settings: SeqD/RCD variants use a sequence length of 16,384; and Chat use 512 tokens for standard tasks and 1,024 for AIME.

| Model | Variant | GSM8K[†] | MATH500 | AIME24 | AIME25 |
|---|---|---|---|---|---|
| SDAR-4B-b32 | Chat[‡] | **86.13** | 50.20 | 5.83 | 2.50 |
| | SeqD | 81.73 | 61.20 | 6.04 | 11.88 |
| | RCD | 84.91 | **65.40** | **9.17** | **17.08** |
| SDAR-4B-b64 | Chat[‡] | 85.90 | 49.80 | 6.25 | 1.67 |
| | SeqD | 78.85 | 56.80 | 4.17 | 7.29 |
| | RCD | **87.04** | **68.40** | **11.04** | **14.79** |
| SDAR-8B-b32 | Chat[‡] | 88.40 | 50.00 | 6.46 | 4.17 |
| | SeqD | 86.50 | 65.80 | 11.67 | 14.79 |
| | RCD | **90.45** | **76.20** | **18.96** | **20.00** |
| SDAR-8B-b64 | Chat[‡] | **88.32** | 51.60 | 5.20 | 2.50 |
| | SeqD | 82.87 | 64.20 | 7.08 | 9.79 |
| | RCD | 86.05 | **74.40** | **18.75** | **16.04** |

[†] We observed potential data contamination in the original Chat models on GSM8K, leading to an inflated baseline that surpasses the SeqD/RCD variants. See Appendix D.

[‡] Chat variants are instruction-following models, whereas SeqD/RCD are further adapted for mathematical reasoning.

*Table 2.* Performance of LLaDA on GSM8K and MinervaMath. We evaluate RCD against the Base and SeqD versions of LLaDA. Here, we use a sequence length of 512 with single-token-per-step decoding. RCD achieves consistent improvements, particularly on the MinervaMath benchmark, where it realizes a nearly 6% absolute accuracy gain, demonstrating the effectiveness of residual context in global-attention diffusion models.

| Model | Variant | GSM8K | MinervaMath |
|---|---|---|---|
| LLaDA | Base[*] | 70.30 | 31.40 |
| | SeqD | 75.74 | 31.10 |
| | RCD | **78.09** | **37.00** |

[*] Results for LLaDA-Base are from the original paper (Nie et al., 2025).

*Table 3.* Throughput-matched accuracy comparison. We use Fast-dllm for LLaDA and D2F for SDAR. By aligning the throughput (Token per Second) of RCD and Sequential Denoising baselines, RCD achieves superior accuracy across all settings.

| Dataset | Variant | Block | Len | TPSec | Acc. (%) |
|---|---|---|---|---|---|
| LLaDA Model (Inference via Fastdllm) | | | | | |
| Minerva Math | SeqD | 64 | 512 | **117.82** | 34.20 |
| | RCD | 64 | 512 | 110.37 | **36.22** |
| GSM8K | SeqD | 64 | 512 | **75.00** | 76.12 |
| | RCD | 64 | 512 | 65.56 | **78.70** |
| SDAR Model (Inference via D2F) | | | | | |
| Minerva Math | SeqD | 64 | 16384 | **130.54** | 50.82 |
| | RCD | 64 | 16384 | 124.86 | **59.82** |
| GSM8K | SeqD | 64 | 16384 | **149.45** | 75.36 |
| | RCD | 64 | 16384 | 148.41 | **81.43** |

*Table 4.* Comparison of RCD and Loopholing under a constrained training budget. Both methods are trained on SDAR-4B-b64 for a single epoch (~300M tokens). "NA" indicates that the model failed to generate valid, evaluatable mathematical sequences. RCD exhibits significantly higher data efficiency, reaching high reasoning accuracy where previous latent-based diffusion methods struggle to achieve basic readability.

| Model | Variant | GSM8K | MATH500 |
|---|---|---|---|
| SDAR-4B-b64 | Loopholing | NA | NA |
| | RCD (1 Epoch) | **86.35** | **66.20** |

tional model for global-context denoising; and the SDAR family (Cheng et al., 2025), a semi-autoregressive model family that decodes sequences in blocks (e.g., $b = 32$ or $b = 64$) and supports KV-cache reuse. For LLaDA, we fine-tune the open-source *base* model into an instruction-following model, where we use a 1M-sample subset of OpenMathInstruct-2 (Toshniwal et al., 2024) (~400M tokens). Since SDAR only provides *chat* versions, we further fine-tune these chat checkpoints into reasoning-specialized models. Here, we use OpenR1-Math-220k (Hugging Face, 2025), filtering for long-context samples (≥8K), totaling ~300M tokens.

**Configurations.** We adopt a standard SFT checkpoint as the reference model ($\mathcal{M}_{\text{ref}}$) for all settings. For SDAR, we employ a *Small-to-Large* strategy where a 1.7B model serves as $\mathcal{M}_{\text{ref}}$ to guide 4B and 8B target models ($\mathcal{M}_{\text{target}}$). We test different block sizes (b32 and b64, representing 32 and 64 tokens per denoising step) to validate RCD's robustness across varying generation granularities. All RCD

models are trained for 5 epochs, with baseline SFT models trained on identical token counts to ensure a fair comparison.

**Evaluation Benchmarks.** We assess mathematical reasoning depth and efficiency across a scaling suite of benchmarks. LLaDA is evaluated on GSM8K (Cobbe et al., 2021) and MinervaMath (Lewkowycz et al., 2022). For the more capable SDAR variants, we extend the evaluation from GSM8K and MATH500 (Hendrycks et al., 2021) to competition-level challenges, including AIME24 (Jia, 2024) and AIME25 (math ai, 2025), to test RCD's efficacy in complex, multi-step logical derivation.

**Evaluation Settings.** To ensure a fairness, we tailor configurations to different settings. (1) Sequence Length: SDAR is evaluated at 16,384 tokens to leverage its reasoning capability. LLaDA uses 512 tokens for standard tasks and 1,024 for AIME, aligning with its training bounds. (2) Decoding: We employ greedy decoding ($T = 0$) for GSM8K, MATH500, and MinervaMath. For AIME24/25, we use $T = 0.6$ with 16-sample Pass@1 to ensure statistical robustness. (3) Residual Initialization: For SDAR, the residual context is initialized with zero vector (cold start), while for LLaDA it is initialized with reference model predictions (warm start).

**Main Results.** As summarized in Tables 1 and 2, RCD consistently outperforms standard Sequential Denoising across all configurations. Most notably, for SDAR-8B-b64, RCD more than doubles accuracy on the rigorous AIME24 benchmark (7.08%→18.75%) and delivers substantial gains on AIME25 (9.79%→16.04%). Similarly, LLaDA shows significant gains, with MinervaMath accuracy increasing by nearly 6%. These results demonstrate RCD's unique ability to recover discarded signals, thereby transforming them into critical contextual priors that significantly enhance reasoning depth.

## 4.2. Efficiency of RCD

**Pareto Frontier Analysis.** To evaluate the efficiency of RCD, we plot accuracy against Token per Step by varying confidence thresholds (0.5–1.0) for both RCD and the Sequential Denoising baseline (Fig. 4). Since RCD introduces minimal computational overhead per step, Token per Step serves as a direct proxy for generation parallelism. Across all model scales and tasks, RCD consistently achieves a superior Pareto frontier, yielding up to a 4–5× computation saving. Specifically, RCD maintains higher accuracy than the one-token-per-step Sequential Denoising baseline even when generating 5 or more tokens per step, effectively reducing the number of decoding iterations.

**Throughput in Practical Frameworks.** We evaluate the deployment efficiency of RCD by integrating it into state-of-the-art inference engines: Fastdllm (Wu et al., 2025c)

for LLaDA (8×H100 GPUs, batch size 4); and D2F (Wang et al., 2025c) for SDAR (8×H100 GPUs, batch size 1). As shown in Table 3, RCD maintains a throughput (Tokens per Second) almost comparable to the baseline while consistently improving accuracy by 2–9% across various models and benchmarks.

## 4.3. Ablation studies

**Interpolation Strategy.** We ablate the choice of $\alpha_i^{(t)}$ in Eq. (2) to identify the optimal interpolation strategy. Beyond our proposed normalized entropy interpolation ($\alpha_i^{(t)} = \hat{H}(\hat{x}_i^{(t)})$), we evaluate fixed-scalar ($\alpha = 0.3$), confidence-based ($\alpha_i^{(t)} = \max(\hat{x}_i^{(t)})$), standard residual ($\alpha = 1.0$), top-1 token (replacing the full predictive distribution with only the most probable token as the residual), and inverse-mapping ($1 - \hat{H}(\hat{x}_i^{(t)}), 1 - \max(\hat{x}_i^{(t)})$) variants. To isolate the efficacy of context extraction from training interference, we conduct evaluations using an untrained SDAR-1.7B-b64 draft model to guide an SDAR-8B-b64 target model at every denoising step under a fixed 0.85 confidence threshold. Results in Fig. 5 demonstrate that normalized entropy achieves the superior Pareto frontier, maximizing parallel decoding speed (Token per Step) while maintaining high reasoning accuracy. Although a marginal accuracy gap exists relative to the baseline in this setting, it is effectively eliminated after SFT. Thus, we adopt the adaptive entropy-based $\alpha$ for its robust information extraction.

**Residual Initialization.** Here, we ablate the initialization of the residual context. Our alternatives are: initializing from a zero vector, from a Dirichlet distribution, from the lightweight reference model's prediction, and from an extra forward pass of the target model itself. As shown in Table 5, all single-model variants outperform the ref-based approach, with Dirichlet initialization achieving the best accuracy (77.4%). Remarkably, even the naive Dirichlet prior, a non-informative distribution, surpasses both the 1.7B and 4B reference models, indicating that low-quality signals from small reference models can actively mislead generation. We leave the exploration of richer initialization schemes to future work.

## 4.4. Discussion

**Scalability across Model and Block Sizes.** RCD demonstrates robust scalability across model parameters and block sizes (Table 1), with performance gains of 4–11 percentage points when scaling from 4B to 8B. Notably, this margin expands as block size $b$ increases, for the simple reason that larger blocks contain more abundant, stabilized contextual prior. Most importantly, the advantage of RCD over other latent methods and looping transformers is its scalability to 8B+ models. Modern 8B+ models use separate parameters for input embedding lagyer and LM head, causing hidden

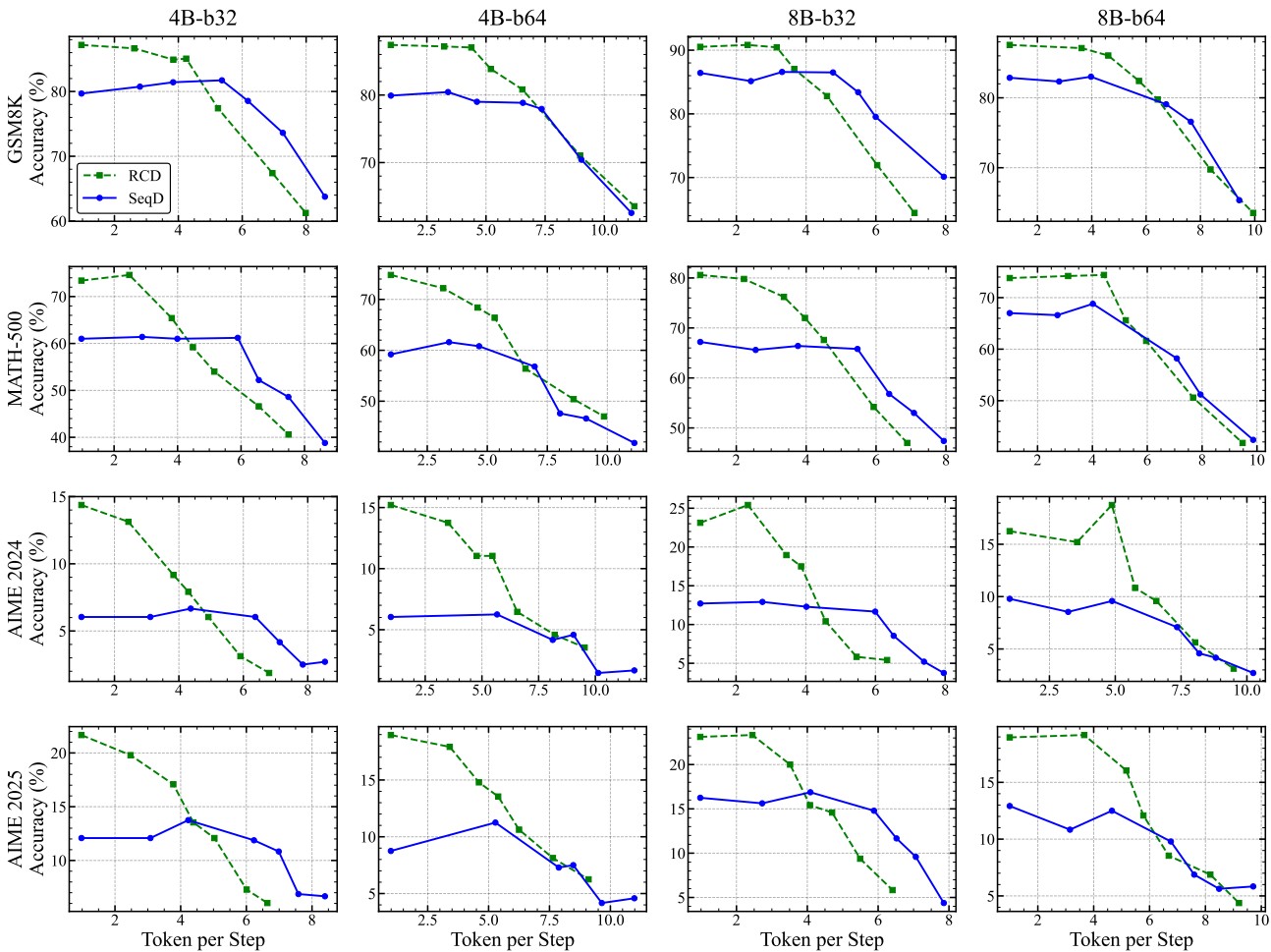

*Figure 4.* Accuracy vs. Token per Step Pareto frontiers for SDAR models. Curves are generated by sweeping confidence thresholds from 0.5 to 1.0. Token per Step measures the average number of tokens generated in each parallel iteration; higher Token per Step indicates fewer total steps for a given length. RCD (green dashed) consistently shifts the frontier toward the upper-left, providing significant speedups at identical accuracy levels. Note that curves may not fully overlap on the x-axis as RCD enables stable generation at higher Token per Step regimes where the Sequential Denoising baseline fails to maintain coherence.

*Table 5.* Initialization strategy ablation on MATH500 (finetuned SDAR-8B-b64 at confidence 0.85). We compare different strategies for initializing the residual context prior distribution. Dirichlet initialization eliminates the reference model while achieving the best accuracy.

| Method | Requires $M_{ref}$? | MATH500 |
|---|---|---|
| Baseline SeqD | No | 64.2 |
| RCD w/ Ref Model (1.7B) | Yes | 74.2 |
| RCD w/ Ref Model (4B) | Yes | 71.8 |
| RCD w/ Target Model (Self) | No | 76.4 |
| RCD w/ Zero Init | No | 75.6 |
| RCD w/ Dirichlet Init | No | **77.4** |

*Table 6.* Saturation analysis on SDAR-8B-b64. RCD (5 epochs) is compared against an extended SeqD baseline (8 epochs). RCD significantly surpasses the saturated baseline, confirming that residual context usage provides gains beyond simple continual training.

| Model | Variant | GSM8K | MATH500 |
|---|---|---|---|
| SDAR-8B-b64 | SeqD (Extended) | 84.61 | 68.00 |
| | RCD | **86.05** | **74.40** |

| Model | Variant | AIME24 | AIME25 |
|---|---|---|---|
| SDAR-8B-b64 | SeqD (Extended) | 8.96 | 10.83 |
| | RCD | **18.75** | **16.04** |

states and input embeddings to reside in divergent semantic spaces. RCD bypasses this bottlenecks by using the model's own input embeddings to construct a coherent latent context regardless of model scale.

**Comparison with Loopholing.** We further compare RCD with Loopholing (Jo et al., 2025), the state-of-the-art latent method for discrete diffusion. The main difference is that Loopholing relies on raw hidden states to transmit infor-

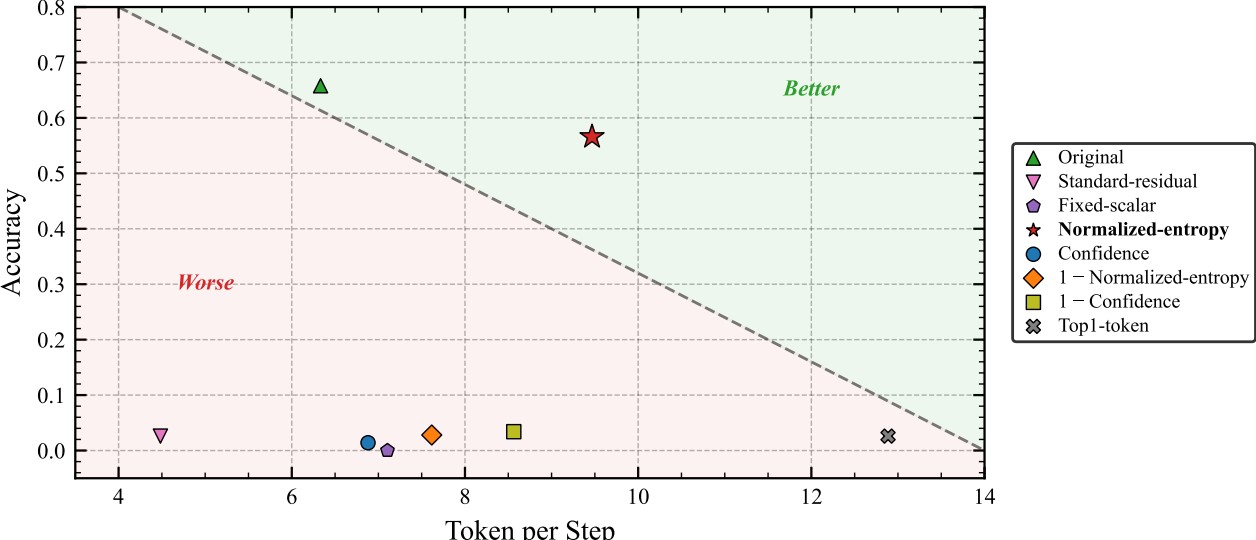

*Figure 5.* Ablation on the interpolation strategy $\alpha_i^{(t)}$ in Eq. (2), evaluated with SDAR-1.7B-b64 as the draft model and SDAR-8B-b64 as the target. We compare eight variants: (1) Normalized Entropy (Ours): $\alpha_i^{(t)} = \hat{H}(\hat{x}_i^{(t)})$; (2) Confidence: $\alpha_i^{(t)} = \max(\hat{x}_i^{(t)})$; (3) Fixed Scalar: $\alpha = 0.3$; (4) Standard Residual: $\alpha = 1.0$; (5) Top-1 Token: using only the most probable token as the residual signal; (6–7) Inverse: $1 - \hat{H}(\hat{x}_i^{(t)})$ and $1 - \max(\hat{x}_i^{(t)})$; and (8) Original: standard SeqD decoding without residual context. Normalized entropy achieves the optimal Pareto frontier, maximizing Token per Step while preserving accuracy. Standard residual and top-1 token both suffer from severe performance degradation, confirming the necessity of adaptive entropy-based weighting and distributional richness.

mation. However, we emphasize that hidden states often exhibit significantly larger norms than input embeddings; injecting them into the input pipeline can destabilize training and inference, overwhelming the original input sequence. We train both RCD and Loopholing on SDAR-4B-b64 using the OpenR1-Math-220k dataset for 1 epoch. As shown in Table 4, RCD adapts rapidly to latent inference, achieving near-optimal reasoning performance on GSM8K and MATH500 within this limited training budget, while the Loopholing counterpart fails to generate coherent sequences under identical conditions. This validates that our proposed method successfully avoids this pitfall by using the model's own input embeddings to ensure numerical stability.

**Training Efficiency and Cost.** RCD achieves rapid convergence with a modest budget of 300M tokens, reaching near-optimal performance in a single epoch (Table 4). This efficiency demonstrates its ability to quickly capture and convert structured contextual signals into reasoning capabilities. While RCD introduces auxiliary training costs (reference model training and inference), the total cost remains practical. For an 8B target and 1.7B reference model, the overhead is within 50%. To test whether baseline performance could be improved by additional training, we extend Sequential Denoising SFT by 60% (3 extra epochs). In Table 6, improvement is modest compared with its 5-epoch counterpart, whereas RCD significantly elevates the accuracy ceiling. This discrepancy suggests the bottleneck in

standard dLLMs is information loss from remasking rather than a lack of training steps. Furthermore, pre-existing fine-tuned models can often serve as reference model, potentially omitting initial reference model training.

## 5. Conclusion

We introduce Residual Context Diffusion, a novel decoding framework that repurposes discarded signals in dLLMs as a structured contextual prior. By introducing entropy to dynamically weight these injected residuals, RCD significantly improves denoising accuracy. Empirical results across benchmarks demonstrate that RCD consistently outperforms standard baselines with comparable inference throughput. We also highlight the scalability of RCD, establishing it as a practical and robust solution for advanced high-fidelity parallel text generation method.

## Acknowledgements

We acknowledge the gracious support from the Furiosa AI, Intel, Apple, NVIDIA, Macronix, and Mozilla team. Furthermore, we appreciate the support from Google Cloud, the Google TRC team Prof. David Patterson, along with support from Google Gemini team, and Divy Thakkar. Prof. Keutzer's lab is sponsored by the Intel corporation, UC Berkeley oneAPI Center of Excellence, Intel VLAB team, as well as funding through BDD and BAIR. We

also acknowledge support by the Director, Office of Science, Office of Advanced Scientific Computing Research, of the U.S. Department of Energy under Contract No. DE-AC02-05CH11231. MWM would also like to acknowledge DARPA, DOE, NSF, and ONR. DOE SciGPT grant. Our conclusions do not necessarily reflect the position or the policy of our sponsors, and no official endorsement should be inferred.

## Impact Statement

This paper presents work whose goal is to advance the field of Machine Learning. There are many potential societal consequences of our work, none which we feel must be specifically highlighted here.

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

# A. Related Works

**Latent Reasoning.** Latent reasoning aims at compacting multiple tokens into one token's embedding vector. This achieves the compression of multiple candidate answers or multiple tokens from the same reasoning step into a single embedding vector, thereby improving the efficiency and accuracy of reasoning. Early methods mainly focus on continuous latent space optimization, i.e., to compress explicit CoT into latent space using techniques such as supervised finetuning and self-distillation given specific reasoning tasks (Hao et al., 2025; Shen et al., 2025; Tan et al., 2025; Su et al., 2025; Ruan et al., 2025; Sun et al., 2025b). Specifically, Zhang et al. (2025b) introduces probability-weighted token embeddings to form a continuous concept space, enabling smooth transitions between abstract concepts. Subsequent research further expanded the application space by employing reinforcement learning for optimization (Li et al., 2025a; Sun et al., 2025a; Li et al., 2025b). Multimodal latent fusion extends reasoning to non-textual domains (Sun et al., 2025a; Bigverdi et al., 2024; Wang et al., 2025a; Zhang et al., 2025a; Yang et al., 2025; Yu et al., 2025b). Although these works demonstrate that latent reasoning can improve computational efficiency and model robustness in complex tasks, two key drawbacks exist. First, once a latent token is generated, it cannot be modified, making backtracking reasoning and contextual refinement challenging. Second, the sequential modeling approach generates only one token at a time containing limited information, resulting in inefficient information propagation and preventing the full context from being incorporated into the latent space within a single forward pass.

**Looped Transformers.** Looped transformers (Merrill & Sabharwal, 2025; Gatmiry et al., 2024; Yu et al., 2025a; Saunshi et al., 2025; Fan et al., 2025; Yang et al., 2024; Xu & Sato, 2025; Yu et al., 2025a; Wu et al., 2025a) leverage cyclic transformer layers to embed iterative reasoning directly into model architecture, avoiding the sequential bottlenecks of chain-of-thought (CoT) prompting. They achieve test-time efficiency by reusing weights across iterative steps, enabling deeper effective computation with minimal parameter overhead. State-of-the-art studies (Zhu et al., 2025c) demonstrate that looped transformers trained on comparable tokens could achieve SOTA (e.g., 2.6B Ouro matches 12B SOTA LLMs) via latent-space iterative computation. However, when the model size is fixed, looped transformers require multiple forward steps to generate a single token compared to the original model. This fundamentally limits the throughput of looped transformers.

**Diffusion Large Language Models.** Masked dLLMs adopt a modeling approach distinct from conventional sequential models, namely masked language modeling. Similar to traditional image diffusion models, they treat text generation as a series of progressive denoising steps. Frontier diffusion dLLMs are categorized into two types: bidirectional dLLMs and block-wise dLLMs. Bidirectional dLLMs, represented by LLaDA (Nie et al., 2025; Zhu et al., 2025b;a; You et al., 2025; Bie et al., 2025) and Dream (Ye et al., 2025), denoise the entire sequence simultaneously. Because the model needs to compute the whole sequence at each denoising step, these models have lower computational efficiency. To address this, block-wise dLLMs partition the sequence into chunks. Early studies following this line aims at improving the inference efficiency of existing bidirectional dLLMs (Wu et al., 2025c; Wang et al., 2025c). Later studies pretrain or finetune the dLLM into a block wise pattern to naturally fit this paradigm (Wu et al., 2025b; Cheng et al., 2025). Other research directions include using model distillation to reduce the number of denoising steps in dLLMs (Deschenaux & Gulcehre, 2025) and training dLLMs' reasoning capabilities using reinforcement learning (Wang et al., 2025d; Zhao et al., 2025).

# B. Formulation of Residual Context Diffusion

**RCD Training.** Algorithm 1 describes the decoupled training framework, where a frozen reference model provides proxy residual signals to the target model.

**RCD Inference.** Algorithm 2 describes the recursive denoising process, incorporating the temperature-adjusted entropy alignment to bridge the distribution gap.

# C. Detailed Training Configurations

In this section, we provide the specific hyperparameter settings used for fine-tuning the SDAR and LLaDA model families. See Table 7 for details.

### C.1. SDAR Family Configuration

The SDAR models (1.7B, 4B, and 8B) were fine-tuned on the OpenR1-Math-220k dataset. To support complex reasoning, we filtered the dataset to retain samples with reasoning chains $\leq$ 8K tokens. The training uses a constant learning rate to

---

**Algorithm 1** RCD Training

---

**input** Training corpus $\mathcal{D}$, pre-trained dLLM $\mathcal{M}_{\text{base}}$, embedding codebook $\boldsymbol{E}$
**output** Trained target model $\mathcal{M}_{\text{target}}$ with parameters $\theta_{\text{target}}$
 1: Initialize $\mathcal{M}_{\text{ref}} \leftarrow \text{SFT}(\mathcal{M}_{\text{base}}, \mathcal{D})$ using standard masked objective
 2: Initialize $\mathcal{M}_{\text{target}} \leftarrow \mathcal{M}_{\text{base}}$
 3: Freeze $\mathcal{M}_{\text{ref}}$
 4: **while** not converged **do**
 5:      Sample $\boldsymbol{x}^{(0)} \sim \mathcal{D}$ and time step $t \sim \mathcal{U}(0, 1)$
 6:      Construct noisy input $\boldsymbol{x}^{(t)}$ by replacing tokens with [M] based on $t$
 7:      $\boldsymbol{p}^{(t)} \leftarrow \mathcal{M}_{\text{ref}}(\boldsymbol{x}^{(t)})$ {Generate reference proxy signal}
 8:      **for** each position $i \in \{1, \ldots, b\}$ **do**
 9:          $\alpha_i^{(t)} \leftarrow \frac{-\sum_{j=1}^{V} p_{i,j}^{(t)} \log p_{i,j}^{(t)}}{\log V}$ {Normalized Entropy}
10:          $\Delta_i^{(t)} \leftarrow \sum_{j=1}^{V} p_{i,j}^{(t)} \boldsymbol{E}_{j,:}$ {Residual Information}
11:          **if** $x_i^{(t)} = [M]$ **then**
12:              $\boldsymbol{e}_i^{(t)} \leftarrow (1 - \alpha_i^{(t)}) E(x_i^{(t)}) + \alpha_i^{(t)} \Delta_i^{(t)}$
13:          **else**
14:              $\boldsymbol{e}_i^{(t)} \leftarrow E(x_i^{(t)})$
15:          **end if**
16:      **end for**
17:      $\mathcal{L} \leftarrow \mathbb{E}[\frac{1}{t} \sum_{i:m_i=1} -\log P_{\theta_{\text{target}}}(\hat{x}_i = x_i^{(0)} \mid \{\boldsymbol{e}_{i'}^{(t)}\}_{i'=1}^{b})]$
18:      Update $\theta_{\text{target}}$ via backpropagation
19: **end while**

---

*Table 7.* Hyperparameters for SDAR and LLaDA Fine-tuning.

| Hyperparameter | Value |
|---|---|
| *SDAR Family Configuration* | |
| Dataset | OpenR1-Math-220k (Filtered $\leq$ 8K) |
| Optimizer | AdamW ($\beta_1 = 0.9, \beta_2 = 0.999$) |
| Learning Rate | $1.0 \times 10^{-5}$ |
| LR Scheduler | `constant_with_warmup` |
| Warmup Ratio | 0.03 |
| Batch Size (per device) | 1 |
| Gradient Accumulation Steps | 12 |
| Total Batch Size | 96 (on 8×H100) |
| Max Sequence Length | 8,192 |
| Training Epochs | 10 (Reference 1.7B) / 5 (Target 4B/8B) |
| Precision | bfloat16 |
| *LLaDA Configuration* | |
| Dataset | OpenMathInstruct-2 (1M subset) |
| Optimizer | AdamW |
| Learning Rate | $1.0 \times 10^{-5}$ |
| LR Scheduler | `cosine` |
| Max Sequence Length | 2,048 |
| Training Epochs | 5 |
| Batch Size (per device) | 2 |
| Gradient Accumulation Steps | 48 |
| Total Batch Size | 768 (on 8×H100) |
| Precision | bfloat16 |
| Distributed Strategy | FSDP |

ensure stability across different model scales.

---

**Algorithm 2** RCD Inference

---

**input** Target model $\mathcal{M}_{\text{target}}$, ref model $\mathcal{M}_{\text{ref}}$, block size $b$, steps $K$, temperature $T_{\text{res}}$, vocabulary size $V$
**output** Denoised sequence $\boldsymbol{x}^{(0)}$

1: Initialize $\boldsymbol{x}^{(t_1)} \leftarrow \{[\text{M}]\}^b$
2: $\boldsymbol{p}^{(t_0)} \leftarrow \boldsymbol{0}$ {Cold start initialization} or $\mathcal{M}_{\text{ref}}(\boldsymbol{x}^{(t_0)})$ {Warm start initialization}
3: **for** each position $i$ **do**
4:    $\alpha_i^{(t_0)} \leftarrow \frac{-\sum_{j=1}^{V} p_{i,j}^{(t_0)} \log p_{i,j}^{(t_0)}}{\log V}$ {Initializing residual states}
5:    $\Delta_i^{(t_0)} \leftarrow \sum_{j=1}^{V} p_{i,j}^{(t_0)} \boldsymbol{E}_{j,:}$ {Initializing residual states}
6: **end for**
7: **for** $k = 1$ **to** $K$ **do**
8:    **for** each position $i$ **do**
9:      **if** $x_i^{(t_k)} = [\text{M}]$ **then**
10:        $\boldsymbol{e}_i^{(t_k)} \leftarrow (1 - \alpha_i^{(t_{k-1})}) E(x_i^{(t_k)}) + \alpha_i^{(t_{k-1})} \Delta_i^{(t_{k-1})}$
11:      **else**
12:        $\boldsymbol{e}_i^{(t_k)} \leftarrow E(x_i^{(t_k)})$
13:      **end if**
14:    **end for**
15:    $\boldsymbol{p}^{(t_k)}, \boldsymbol{z}^{(t_k)} \leftarrow \mathcal{M}_{\text{target}}(\{\boldsymbol{e}_i^{(t_k)}\}_{i=1}^{b})$
16:    $c_i^{(t_k)} \leftarrow \max(\boldsymbol{p}_i^{(t_k)})$ for all masked positions
17:    $S \leftarrow$ top-$m$ positions with highest confidence $c_i^{(t_k)}$
18:    **for** each $i \in S$ **do**
19:      Sample $\hat{x}_i \sim \boldsymbol{p}_i^{(t_k)}$ and set $x_i^{(t_{k+1})} \leftarrow \hat{x}_i$
20:    **end for**
21:    Set $x_i^{(t_{k+1})} \leftarrow x_i^{(t_k)}$ for all $i \notin S$ {Remasking}
22:    **for** each position $i$ **do**
23:      $p_{i,j}^{(t_k)}(T_{\text{res}}) \leftarrow \frac{\exp(z_{i,j}^{(t_k)}/T_{\text{res}})}{\sum_{j'=1}^{V} \exp(z_{i,j'}^{(t_k)}/T_{\text{res}})}$
24:      $\alpha_i^{(t_k)} \leftarrow \frac{-\sum_{j=1}^{V} p_{i,j}^{(t_k)}(T_{\text{res}}) \log p_{i,j}^{(t_k)}(T_{\text{res}})}{\log V}$ {Updating residual states}
25:      $\Delta_i^{(t_k)} \leftarrow \sum_{j=1}^{V} p_{i,j}^{(t_k)} \boldsymbol{E}_{j,:}$ {Updating residual states}
26:    **end for**
27: **end for**

---

## C.2. LLaDA Configuration

The LLaDA-8B-Base model was fine-tuned on a 1M subset of the OpenMathInstruct-2 dataset using FSDP for distributed training. Unlike the block-wise SDAR, LLaDA uses a larger context window of 2048 tokens and a standard SFT paradigm to optimize its global bidirectional context.

## D. Potential Data Contamination on GSM8K

In our main experiments (Table 1), we observe a paradoxical phenomenon where the SDAR-Chat models, which are general instruction-following models, occasionally outperform our reasoning-specialized Sequential Denoising and RCD models on the GSM8K benchmark. To investigate the reason, we conduct a cross-benchmark evaluation using GSM1K (Zhang et al., 2024) and GSM-Plus (Li et al., 2024). Among these benchmarks, GSM1K consists of newly curated problems designed to mimic the difficulty of GSM8K but with zero possibility of appearing in existing training corpora. GSM-Plus introduces adversarial perturbations (e.g., numerical changes, irrelevant information) to original GSM8K problems to test robust generalization.

As shown in Table 8, while the SDAR-Chat models achieve high scores on the original GSM8K (up to 88.40%), their performance degrades significantly on more robust benchmarks. On GSM1K, we observe a consistent drop of 2–7%, suggesting that the models rely partially on seen patterns from the training distribution. On **GSM-Plus**, the performance

*Table 8.* Accuracy of SDAR-Chat models across three GSM variants. The significant performance drop from GSM8K to GSM1K/Plus indicates potential data contamination.

| Model | Block | GSM8K | GSM1K | GSM-Plus |
|---|---|---|---|---|
| SDAR-4B-Chat | 32 | 86.13 | 82.57 | 75.89 |
| | 64 | 85.90 | 79.25 | 73.87 |
| SDAR-8B-Chat | 32 | 88.40 | 86.22 | 78.38 |
| | 64 | 88.32 | 83.24 | 77.62 |

plunges by over 10% across all configurations. This indicates that the high GSM8K scores are not representative of robust mathematical reasoning; rather, they reflect a fragile memorization that fails when problem surface features are even slightly perturbed.

