# OpenReview forum: "Residual Context Diffusion Language Models"
_ICML.cc/2026/Conference — ICML 2026 regular_

### Official Review · Reviewer_xVy7 · 2026-03-05

**Soundness:** 3
**Presentation:** 3
**Significance:** 3
**Originality:** 3
**Overall Recommendation:** 4
**Confidence:** 4

**Summary:**

Diffusion Large Language Models (dLLMs) have emerged as a promising alternative to purely autoregressive (AR) models due to their ability to decode multiple tokens in parallel, demonstrating encouraging results in tasks such as instruction following, code generation, long-context understanding, and complex reasoning. However, current state-of-the-art dLLMs rely on a remasking strategy that discards tokens potentially containing useful information rather than effectively utilizing them, thereby wasting computation. To address this, the paper proposes Residual Context Diffusion (RCD), which integrates the information carried by previously discarded tokens into the next denoising step. Furthermore, the paper introduces a decoupled two-stage training pipeline that leverages a lightweight reference model to alleviate the memory bottlenecks associated with backpropagation in large target models. Extensive experiments on mathematical reasoning tasks validate the effectiveness of the proposed method.

**Compliance With Llm Reviewing Policy:**

Affirmed.

**Final Justification:**

Considering the overall contribution and quality of this work, I maintain my judgment of weak accept, but certain revisions need to be made in the final version based on the rebuttal content.

**Key Questions For Authors:**

As discussed in the "Strengths and Weaknesses", I suggest that the authors incorporate the following experimental results and analyses into Section 4:
* Please provide results for the inference pipeline where the warm start mechanism using the reference model (Mref) is disabled (e.g., starting with standard mask embeddings).
* Please include an evaluation of RCD performance when using reference models (Mref) of varying scales.
* Please provide a comparative experiment where only the highest-probability token (top-1) is used as the residual, rather than the full predictive distribution, to justify the claimed benefits of distributional richness.
* Please demonstrate the performance of RCD on broader task domains beyond mathematical reasoning, such as code generation or instruction following.

**Limitations:**

yes

**Strengths And Weaknesses:**

**Strengths**
* The paper identifies the inherent flaws of the remasking strategy in current block-wise dLLMs and introduces RCD. While the concept of leveraging historical context has been explored in other subfields of machine learning, the systematic recycling and integration of discrete probability distributions within the Discrete Diffusion paradigm remains a relatively under-explored and highly challenging direction. RCD provides an intuitive and effective path to enhance both the reasoning depth and logical coherence of dLLMs.
* To address the memory explosion and backpropagation bottlenecks encountered by LLMs during recursive denoising training, the paper proposes a decoupled two-stage training pipeline. This method introduces a lightweight reference model to provide stable proxy residual signals, thereby effectively circumventing the computational burden associated with recursive training.
* The overall narrative of the paper is fluid, and the logical transitions are seamless.

**Weaknesses**
* In the inference pipeline, the authors utilize a reference model (Mref) to perform "warm up" at the very first denoising step (t1) because no previous residual exists. However, it remains unclear to what extent the reported performance improvements originate from this high-quality "anchor" versus the subsequent recursive decoding process. Section 4 fails to provide ablation studies or comparative results to decouple these two potential sources of gain.
* In the configurations described in Section 4, the authors employ a "Small-to-Large" strategy where a 1.7B model serves as Mref to guide 4B and 8B target models (Mtarget). Nevertheless, it remains unexplored whether the performance of RCD is sensitive to the scale of the reference model or its fine-tuning status.
* In Section 1, the authors state that "it is unclear where residual contextual signals should come from and how they should be aggregated." However, Section 4 does not provide an empirical comparison between feeding back "the token prediction itself" (hard tokens) versus the "full predictive distribution" (soft tokens).
* The experimental evaluations and analyses in Section 4 are exclusively conducted on mathematical reasoning tasks, specifically GSM8K, MATH500, and AIME. While the performance gains in this domain are notable, the diversity of task types is quite limited. Since mathematical reasoning is highly structured, it remains unproven whether the proposed RCD can maintain its effectiveness or stability in other critical areas, such as Code Generation or Instruction Following, which may present different linguistic patterns and reasoning requirements.

---

> ### Author Rebuttal · Authors · 2026-03-27
>
> Dear Reviewer xVy7,
>
> Thank you for recognizing the potential and effectiveness of our work and for providing detailed constructive comments.
>
> Below, we address each point raised.
>
> > **Q1:** Please provide results for the inference pipeline where the warm start mechanism using the reference model (M_ref) is disabled (e.g., starting with standard mask embeddings).
>
> **R1:** Here, we completely removed the ref model and evaluated alternative initialization strategies.
>
> As shown below, initializing with the M_target itself ("Self"), zero, or a Dirichlet distribution eliminates all extra parameters while actually **outperforming** the ref-based approach. All single-model variants maintain a significant lead over the SeqD baseline. This reveals that the success of RCD is **not dictated by the quality or size of the ref model**. Instead, the system relies on its internal residual context mechanism, and M_ref is **not strictly necessary** for our method.
>
> **Table. Accuracy on MATH500 (finetuned SDAR-8B-b64).**
>
> | Method (Init Strategy)     | Requires M_ref? | MATH500                        |
> | -------------------------- | --------------- | ------------------------------ |
> | Baseline SeqD              | ✖               | 64.2                           |
> | RCD w/ Ref Model           | ✔               | 74.2 (1.7B ref); 71.8 (4B ref) |
> | RCD w/ Target Model (Self) | ✖               | 76.4                           |
> | RCD w/ Zero Init           | ✖               | 75.6                           |
> | RCD w/ Dirichlet Init      | ✖               | **77.4**                       |
>
> > **Q2:** Please include an evaluation of RCD performance when using reference models (M_ref) of varying scales.
>
> **R2:** To evaluate the reliance on M_ref, we tested a larger 4B ref model as well as the 8B target itself as ref(see Table in R1).
>
> First, regarding standalone performance, the 1.7B and 4B ref models achieve standalone accuracies of **34.0** and **56.8**, respectively. The target model (8B) is considered to have **75.6** accuracy.
>
> Second, results show that the target-as-ref setting outperforms 1.7B and 4B ref settings, indicating that better M_ref potentially does lead to better performance.
>
> However, we’d also like to highlight that 1.7B and 4B reference settings explicitly underperform the naive Dirichlet initialization. This indicates that some **low-quality "misleading" signals from the small ref models could even corrupt the generation quality, causing it to underperform simple non-informative random initialization**. Therefore, the pipeline actually **does not need a "good" M_ref to be successful**.
>
> > **Q3:** Please provide a comparative experiment where only the highest-probability token (top-1) is used as the residual, rather than the full predictive distribution, to justify the claimed benefits of distributional richness.
>
> **R3:** Our ablation study is shown in the updated table below. Replacing the full predictive distribution with top-1 token leads to severe performance degradation. This demonstrates the necessity of aggregating all possible predictions into soft tokens.
>
> **Table: Ablation on Residual Component (Corresponds to Figure 5)**
>
> | Residual Component               | MATH500  |
> | -------------------------------- | -------- |
> | Top-1 token                      | 2.6      |
> | **Full predictive distribution** | **56.6** |
>
> > **Q4:** Please demonstrate the performance of RCD on broader task domains beyond mathematical reasoning, such as code generation or instruction following.
>
> **R4:** Here, we finetune LLaDA model on the Tulu3 [1] dataset and report its performance on MBPP [2] and Hellaswag [3].
>
> This demonstrates RCD's effectiveness on non-mathematical tasks.
>
> |      | **MBPP** | **Hellaswag** |
> | ---- | -------- | ------------- |
> | SeqD | 30.6     | 76.7          |
> | RCD  | **33.4** | **83.9**      |
>
> [1] Tulu 3: Pushing Frontiers in Open Language Model Post-Training, COLM 2025 (arxiv:2411.15124)
>
> [2] Program Synthesis with Large Language Models, CACM (arxiv:2108.07732)
>
> [3] HellaSwag: Can a Machine Really Finish Your Sentence?, ACL 2019 (arxiv:1905.07830)

---

> > ### Author Rebuttal · Reviewer_xVy7 · 2026-04-02
> >
> > Thank the authors for their detailed response and the supplementary experiments provided. The responses to Q3 and Q4 have addressed my concerns, but the results for Q1 and Q2 have raised the following new questions:
> >
> > 1. As shown by the further experiments in R1, Mref did not yield valuable gains and even compromised model performance. Consequently:
> >
> > - a. The authors should carefully consider revising the "Inference Pipeline" part in Section 3.3 of the original paper.
> >
> > - b. If using Dirichlet initialization yields better results than using Mref to provide an "anchor" during the inference phase, shouldn't there have been attempts to use Dirichlet initialization directly instead of Mref during the training phase? If Mref is needed for neither training nor inference, the necessity of the "decoupled two-stage pipeline" described in the paper becomes questionable.

---

> > > ### Author Response · Authors · 2026-04-02
> > >
> > > Dear Reviewer xVy7,
> > >
> > > Thank you for your prompt reply and for acknowledging our previous responses. Your follow-up questions are highly insightful. Here, we address them in detail:
> > >
> > > > **a. The authors should carefully consider revising the "Inference Pipeline" part in Section 3.3 of the original paper.**
> > >
> > > **Response (a):** We fully agree. Based on the new findings from our rebuttal experiments, we will substantially revise Section 3.3 ("Inference Pipeline") in the final manuscript. We will update the single-model method and its performance accordingly. We will also update the corresponding algorithm blocks to include them.
> > >
> > > > **b. If using Dirichlet initialization yields better results... shouldn't there have been attempts to use Dirichlet initialization directly instead of M_ref during the training phase? ... the necessity of the "decoupled two-stage pipeline" becomes questionable.**
> > >
> > > **Response (b):** This is an excellent point that touches on the core of our training design. We wish to clarify that while $M_{ref}$ can be removed during **inference**, it remains absolutely essential during **training**. The decoupled two-stage training pipeline is still necessary, for the following reasons:
> > >
> > > 1. **Simulating Intermediate Residual States During Training:** During inference, the Dirichlet initialization is *only* applied at the very first step (when the sequence is 100% masked). However, during training, the target model must learn to denoise across *various intermediate noise levels*. At these intermediate stages, the input $p$ is no longer pure noise; it contains structured, partially confident residual information. A random Dirichlet distribution cannot simulate these meaningful intermediate soft distributions.
> > > 2. **$M_{ref}$ acts as the Training "Anchor":** To teach the target model how to effectively utilize residual context, we need informative token distributions ($p$) at different masking ratios during training. $M_{ref}$ serves precisely this purpose. **Furthermore, $M_{ref}$ does not need to be a massive model to be effective. During training, the input provided to $M_{ref}$ is constructed by adding noise to the actual ground truth sequence. Because it is conditioned on this ground-truth context, even a smaller $M_{ref}$ can reliably generate high-quality residual distributions to guide the target model.** Without $M_{ref}$ generating these realistic states, the target model would entirely miss the core capability of "reusing residual context." (On the other hand, during inference, for there are no ground truths given, the small reference model appearantly "misleads" the target model.)
> > > 3. **Empirical Dependency:** It is crucial to note that the target model achieving excellent inference results with Dirichlet initialization (as shown in our previous response) was **trained using the decoupled two-stage pipeline with $M_{ref}$**. While random Dirichlet is employed at the 'starting point' (initialization), the success of the 'intermediate stages' (denoising phases) still **relies on** the robust denoising capabilities and residual context utilization learned from Mref during training.
> > >
> > > In summary, Dirichlet initialization is merely an inference-time trigger, while $M_{ref}$ is the indispensable training-time teacher. We will add a dedicated paragraph in our revision to clarify this critical distinction between training necessity and inference flexibility.

---

### Official Review · Reviewer_eec1 · 2026-03-10

**Soundness:** 3
**Presentation:** 3
**Significance:** 3
**Originality:** 3
**Overall Recommendation:** 4
**Confidence:** 4

**Summary:**

This paper studies residual information in diffusion language models: even masked positions that are not yet selected for unmasking can still contain useful soft information. The proposed method, Residual Context Diffusion (RCD), carries this information across denoising steps by computing a weighted embedding from the token distribution at masked positions and injecting it into the next step. This requires a newly trained model, since the input is no longer just masked/unmasked token embeddings, but also includes residual soft embeddings from still-masked positions. The training recipe uses a reference model to provide stable token distributions early in training, which is intended to mitigate instability from unrolling multiple denoising steps. The claimed benefit is that RCD can both reuse information that standard decoding discards and make later denoising steps more informed, which could improve generation quality in addition to potentially reducing decoding cost.

**Compliance With Llm Reviewing Policy:**

Affirmed.

**Final Justification:**

I maintain a positive view of the high-level concept and the strong empirical results presented in the paper (and thereby raised my score). My primary reservation during the initial review was the counterintuitive nature of placing higher weights on high-entropy residuals. Intuitively, this approach seemed to risk amplifying noise rather than preserving useful information.

The authors provided a compelling rebuttal that addressed this concern through two main clarifications. They provided dataset statistics showing that the absolute entropy values in their operating regime are quite low, with a Q3 of 0.1455. This confirms that the model is dealing with moderate uncertainty and meaningful ambiguity rather than pure uniform noise. Then, they drew a critical distinction between hard decoding and soft residual context propagation. By up-weighting soft embeddings, the method preserves a superposition of possibilities that the diffusion model can resolve in subsequent denoising steps, rather than forcing premature discrete errors.

These explanations successfully alleviate my initial concerns regarding the potential for catastrophic noise amplification. However, I still think that the core arguments remain fundamentally heuristic. The empirical success is undeniable, but the theoretical justification still relies heavily on trusting the diffusion language model to sort out these amplified ambiguities later in the process. We lack a rigorous, fundamental understanding as to why prioritizing this ambiguity is strictly optimal compared to focusing on nearly finalized tokens.

Despite this reliance on heuristics, the empirical evidence strongly supports the authors' claims, and the rebuttal provides a logical intuition for the method's success. I am satisfied with the authors' response and view the paper favorably. I recommend that the authors be required to integrate the provided entropy statistics and the discussion on soft propagation into the camera-ready version, as these details are absolutely essential for future readers to fully grasp the mechanics of the approach.

**Key Questions For Authors:**

- What exactly is $m$ in each experiment, and how sensitive are the results to this choice?
- Why should higher-entropy token distributions receive larger residual weights, while the decoding process itself still follows high-confidence-first unmasking?
- Can the authors provide stronger evidence that RCD improves generation quality, not only efficiency? What is the mechanism behind the quality gain?

**Limitations:**

The authors have not discussed the limitation (nor societal impact).
See weaknesses for limitations.

**Strengths And Weaknesses:**

> Strengths

- The paper addresses a reasonable limitation of diffusion LMs, where standard decoding may discard useful uncertainty information too aggressively.
- The core idea is intuitive and technically interesting. Propagating soft residual context instead of relying only on hard unmasking decisions is reasonable.
- The method is not just an inference heuristic, but it includes a dedicated training strategy to support the modified input structure.
- Using a reference model to stabilize early training seems interesting (and practical) idea.
- As the author already showed, this approach appears potentially compatible with fast diffusion decoding engines, which increases its practical relevance.

> Weaknesses

- The central weighting design is not fully convincing. The method appears to give larger influence to high-entropy / low-confidence positions, while the decoding policy still prioritizes high-confidence unmasking.
- The justification for why higher-entropy tokens should contribute more residual signal needs stronger theoretical or empirical support.
- The claimed quality improvement is less obvious than the speedup intuition. It is not trivial that preserving uncertain residual information should consistently improve final performance.
- The method requires training a new model, which may limit ease of adoption compared with plug-in inference-only methods. (I think this is minor issue).
- Some experimental details seem unclear, especially the choice of $m$, the number of positions unmasked at each decoding stage.

---

> ### Author Rebuttal · Authors · 2026-03-27
>
> Dear Reviewer eec1,
>
> Thank you for recognizing the potential and effectiveness of our work and for providing detailed constructive comments. Below, we address each point raised.
>
> > **W1:** The method gives larger influence to high-entropy positions, while the decoding policy prioritizes high-confidence unmasking.
>
> **R1:** While it might seem counterintuitive, these two mechanisms are in fact complementary:
>
> 1. **An Information-Theoretic Perspective:** A token with high entropy naturally possesses more potential information. If effectively propagated, it can better improve the overall decoding confidence of the sequence [1]. Therefore, assigning larger weights to these information-rich tokens is conceptually well-justified. As demonstrated in our ablation study (Figure 5), assigning higher weights to high-entropy tokens provides the best parallelism and maintains accuracy, while alternative strategies completely fail to achieve satisfactory performance.
> 2. **Balancing Robustness and Information Propagation:** Prioritizing high-confidence tokens during decoding is to ensure generation robustness, but it inherently suppresses the propagation of information of high-entropy tokens [2]. We solve this by forwarding them in a much safer manner: we retain them as latent vectors instead of explicitly decoding them. Thus, the decoding policy actually benefits from the information flowing through these high-entropy latent states.
>
> [1] From Bits to Rounds: Parallel Decoding with Exploration for Diffusion Language Models, ICLR, 2026 (arxiv:2511.21103)
>
> [2] The Flexibility Trap: Why Arbitrary Order Limits Reasoning Potential in Diffusion Language Models, arxiv:2601.15165
>
> > **W2:** Stronger theoretical or empirical support for entropy weighted residual signals.
>
> **R2:** Please refer to W1.
>
> > **W3:** Evidence and reason for quality improvement.
>
> **R3:** The most direct evidence is our benchmark results. For instance, on MATH500, RCD achieves performance improvement of **65.80 → 77.60**.
>
> Additionally, we directly evaluate generation quality for AIME24 via ReasonEval [3]:
>
> ||**Validity ↑**|**Redundancy ↓**|
> |-|-|-|
> |SeqD 8B b64|0.80|0.60|
> |RCD 8B b64 (ours)|**0.87**|**0.58**|
>
> We explain the reason in two aspects:
>
> 1. **For individual tokens:** A dLLM transforms masks into specific tokens. If the mask position retains substantial prior information about the target token (rather than being reset to a pure mask embedding), the transform is much smoother, and the difficulty is significantly reduced.
> 2. **For surrounding tokens:** dLLMs operate similarly to solving a "fill-in-the-blank" problem, making them highly dependent on bidirectional context. By infusing the masked positions with retained contextual information, we provide a much more informative global context. This facilitates the decoding of other masked tokens within the same sequence.
>
> [3] Evaluating Mathematical Reasoning Beyond Accuracy, AAAI 2025 (arxiv:2404.05692)
>
> > **W4:** The method requires training a new model.
>
> **R4:** We agree that a plug-and-play method would have better accessibility. However, we want to highlight that RCD only needs lightweight training. Compared with the autoregressive → dLLM transform (50B tokens reported by [4]), our **RCD training requires ~100x less training budget**, with a **minimum of ~300M tokens**.
>
> > **W5:** Choice of m (the number of positions unmasked at each decoding stage)?
>
> **R5:** In our experiments, m is not an independent variable. Instead, we follow confidence decoding, the most widely used decoding method that only tokens more confident than a specific threshold could be decoded. Thus, different m is reached via changing this confidence threshold:
>
> 1. In Table 1, 4, 5 and Figure 5, we mainly evaluate model quality. Thus, we measure the overall performance under a specific confidence level (0.85). In Table 2, the threshold is set to 1.0 (m is 1 in this case), similarly. These align with [4] and [5].
> 2. For the speedup test in Table 3, we tune the threshold to align the generation throughput. The detailed parameters will be updated in the paper.
> 3. In Figure 4, our goal is to measure the efficiency of RCD, so we vary the thresholds in 0.5–1.0 and explicitly plot the m (Token per Step) vs. accuracy as the Pareto Frontier.
>
> As of the sensitivity of m, we observe in Figure 4 that the performance of RCD is relatively stable across different m values (m<4), with a slight decrease in performance as m increases. This proves RCD's robustness to the choice of m.
>
> [4] SDAR: A Synergistic Diffusion-AutoRegression Paradigm for Scalable Sequence Generation, arxiv:2510.06303
>
> [5] dLLM: Simple Diffusion Language Modeling, arxiv: 2602.22661
>
> > **Q1:** Choice of m, and its sensitivity?
>
> **R1:** Please refer to W5.
>
> > **Q2:** Why should higher-entropy token get larger weights?
>
> **R2:** Please refer to W1.
>
> > **Q3:** Stronger evidence that RCD improves generation quality and its mechanism.
>
> **R3:** Please refer to W3.

---

> > ### Author Rebuttal · Reviewer_eec1 · 2026-04-02
> >
> > The rebuttal still does not fully resolve my concern. The argument appears to conflate high entropy with useful information. In my view, high entropy only indicates that the output is uncertain. It does not by itself show that the residual signal is informative rather than noisy. A high-entropy token may correspond to meaningful ambiguity, but it may also simply reflect model confusion. Therefore, the cited evidence seems to support at most that some uncertain tokens can be beneficial to explore, not the stronger claim used here that larger residual weights should be assigned to higher-entropy positions in general.

---

> > > ### Author Response · Authors · 2026-04-06
> > >
> > > Dear Reviewer eec1,
> > >
> > > Thank you for your continued engagement and valuable feedback. We completely understand your core concern: high entropy can often indicate pure model confusion and noise, and assigning larger weights to such noise seems counterintuitive and potentially harmful.
> > >
> > > We appreciate the opportunity to clarify this crucial point. Our choice of method is grounded in the actual entropy distribution of our model, the distinction between hard decoding and soft propagation, and a strict empirical evidence chain. We address your concerns in detail below:
> > >
> > > **1. Meaningful Ambiguity in High Entropy**
> > >
> > > We agree that extremely high entropy (e.g., approaching a uniform distribution) represents pure noise and confusion. However, it is crucial to examine the *actual operating regime* of our method. We analyzed the token entropy distribution during decoding on the MATH500 dataset, and the results show that the absolute entropy values are actually very low:
> > >
> > > | **Metric**  | **Q1** | **Median** | **Q3** |
> > > | ----------- | ------ | ---------- | ------ |
> > > | **Entropy** | 0.0077 | 0.0574     | 0.1455 |
> > >
> > > With a Q3 of only 0.1455, the "high entropy" tokens we assign larger weights to are not pure random noise. Instead, they represent **moderate uncertainty with meaningful ambiguity**. In this specific low-to-moderate entropy regime, as demonstrated in [1], the information gain and contextual value of exploring these tokens far outweigh the minor noise they introduce.
> > >
> > > **2. Soft Propagation vs. Hard Decoding**
> > >
> > > Furthermore, we must emphasize the fundamental difference between standard "hard decoding" and our "soft residual context" propagation.
> > >
> > > If we forced the model to hard-decode these high-entropy positions into discrete tokens, it would indeed introduce errors and hallucinations, exactly as you are concerned about. However, RCD extracts and propagates **soft embeddings**. By assigning higher weights to these soft representations, we are not amplifying wrong, premature decisions. Rather, we are preserving the **superposition of possibilities**(meaningful ambiguity). The diffusion LM's strong denoising capability then uses subsequent steps, alongside a more complete global context, to safely resolve these ambiguities.
> > >
> > > **3. Soft Propagation Increases Internal Confidence**
> > >
> > > To directly address your concern about whether propagating higher-entropy signals disrupts the model's internal states, we conducted a **new supplementary experiment** to analyze the average decoding confidence under different interpolation strategies on MATH500.
> > >
> > > If our method were purely amplifying noise, we would expect the model to become more confused, leading to lower overall confidence. However, our new results show the exact opposite:
> > >
> > > | **Method (Interpolation Strategy)** | **Avg Confidence** |
> > > | ----------------------------------- | ------------------ |
> > > | Linear (Fixed α = 0.3)              | 0.86               |
> > > | Confidence                          | 0.86               |
> > > | Inverse + Confidence                | 0.89               |
> > > | Inverse + Normalized-entropy        | 0.88               |
> > > | **Ours (Normalized-Entropy)**       | **0.93**           |
> > >
> > > By utilizing the Normalized-Entropy strategy, the average confidence *increases* to 0.93. This newly added evidence strongly suggests that the residual signals act as **constructive global context**, helping the model resolve early uncertainties rather than being overwhelmed by noise.
> > >
> > > **4. End-to-End Empirical Validation**
> > >
> > > This increased internal confidence perfectly aligns with our end-to-end performance. As shown in our ablation study (Figure 5), assigning higher weights to high-entropy tokens ("Ours") drastically outperforms both assigning higher weights to low-entropy tokens ("Confidence") and treating all tokens equally ("Linear"):
> > >
> > > | **Method (Interpolation Strategy)** | **MATH500** |
> > > | ----------------------------------- | ----------- |
> > > | Linear (Fixed α = 0.3)              | 0.0         |
> > > | Confidence                          | 1.4         |
> > > | Inverse + Confidence                | 3.4         |
> > > | Inverse + Normalized-entropy        | 2.8         |
> > > | **Ours (Normalized-Entropy)**       | **56.6**    |
> > >
> > > Overall, the dual improvement in both the newly measured average confidence and the final accuracy proves that the model successfully utilizes the residual signals of high-entropy positions as constructive context to resolve ambiguity. We will carefully incorporate this discussion, the supplementary confidence experiment, and the entropy distribution statistics into the final manuscript. Thank you again for providing insightful suggestions on our study!
> > >
> > > [1] From Bits to Rounds: Parallel Decoding with Exploration for Diffusion Language Models, ICLR, 2026 (arXiv:2511.21103)

---

### Official Review · Reviewer_bipp · 2026-03-11

**Soundness:** 3
**Presentation:** 2
**Significance:** 2
**Originality:** 2
**Overall Recommendation:** 5
**Confidence:** 4

**Summary:**

This work aims to improve diffusion language models by reusing model outputs from the previous diffusion iteration, rather than resetting low-confidence tokens to mask tokens, which completely discards prior predictions. The authors propose two methods: (1) integrating previous token probabilities via entropy-based weighting, and (2) adopting a two-stage training procedure to avoid expensive backpropagation through multiple diffusion steps.

**Compliance With Llm Reviewing Policy:**

Affirmed.

**Final Justification:**

I have increased my score from 3 to 5 (and the soundness from 2 to 3), as the authors' responses and additional experimental results provided in the rebuttal have adequately addressed all of my major concerns.

I expect that the authors will take into account all the discussions and comments to improve the final version.

**Key Questions For Authors:**

Please refer to the Weaknesses Section.

**Limitations:**

Please refer to the Weaknesses Section.

**Strengths And Weaknesses:**

# Strengths:

Research aimed at improving diffusion language models is highly relevant to many researchers in the ICML community.

Apart from the issues highlighted in the Weaknesses section below, the paper is overall sound.

# Weaknesses:

I would like to clarify that I am willing to increase the score if all the following concerns are adequately addressed.

**Misleading method name:**

(1) The method is called “Residual Context Diffusion” but in reality, it uses a highway connection [Ref1] (which itself was inspired by LSTM [Ref 2]) instead of the residual connection (see Eq. 2).

If the gating (which is one of the main contributions of this work) is really important as claimed in the paper, the method should be called Highway Context Diffusion instead. Otherwise, the current name is inconsistent with the actual type of skip connection used.

This distinction is not a minor detail given that the novelty of the residual connection was precisely due to the removal of gating in the highway connection.

[Ref 1] Srivastava, Greff, Schmidhuber. Training Very Deep Networks. NeurIPS 2015

[Ref 2] Hochreiter, Schmidhuber. Long Short-Term Memory. Neural Computation 1997

(2) Related to above: Is the proposed/highway version better than the residual version? While the reviewer appreciates the ablation study conducted in Sec 4.3, one ablation study that is missing is to set alpha to 1, corresponding to the true residual connection. I believe “Linear” in Figure 5 uses a fixed alpha but I could not find the actually used value in the main text (sorry if it’s reported somewhere and I overlooked it).

**Experimental setting issues/limitations:**

(3) In the proposed method, even the inference requires two models: M_ref to produce the initial probability distributions and M_target for the rest. This virtually increases the parameter counts of the entire system combining two models, which, strictly speaking, makes the comparison unfair to the baseline SeqD using a single model. 1.7B parameters of the reference model is not negligible compared to the target model size of 8B.

(4) Related to above, it is unclear how good M_ref needs to be for the entire pipeline to be successful. How good is M_ref when evaluated as a standalone system? Does using a better/worse M_ref improve/deteriorate the entire system (respectively)?

(5) While I completely understand that the focus of this work is on improving the diffusion language models, it would still be useful to show the reference performance numbers of a comparable auto-regressive language model baseline. This is useful (i) to keep track of how much performance gap still exists between diffusion vs. auto-regressive models, and (ii) to show by how much this gap is reduced by the proposed method.

**Writing/clarity issues:**

(6) The following sentence is misleading: “today’s dLLMs still trail AR models in accuracy. This gap is counterintuitive” (line 38 right, page 1). In the literature of non-autoregressive language models, this performance gap has always been intuitive in light of the probabilistic view, because auto-regressive processing corresponds to a proper decomposition of the sentence probability as a product of the conditional probabilities of each token given all its predecessors. I believe what the authors really meant is not “counterintuitive” but rather “unsatisfactory” given the computational cost of multiple sequential denoising iterations.

(7) Sec 3.2 Step 1 gives the impression that M_target is also initialized using the parameters of M_ref (which, as readers later discover, is not the case; M_target and M_ref can be of different sizes). I found this confusing while reading Sec 3.2 Step 1.

(8) The comment below Table 1 says: “We observed potential data contamination in the original Chat models on GSM8K, leading to an inflated baseline that surpasses the SeqD/RCD variants.” However, given that the Chat baseline is used as the initialization for both SeqD/RCD, it is unclear why the contamination only affects the baseline but not SeqD/RCD.

---

> ### Author Rebuttal · Authors · 2026-03-27
>
> Dear Reviewer bipp,
>
> Thank you for recognizing the potential and effectiveness of our work and for providing detailed constructive comments. Below, we address each point raised.
>
> > **W1:** The current name (Residual Context Diffusion) is inconsistent with the actual type of skip connection used (highway connection).
>
> **R1:** We sincerely thank you for raising this insightful point regarding our terminology. Our initial choice of the word "Residual" was simply to convey the idea of "recycling" discarded context by feeding it forward. In the revised manuscript, we will explicitly incorporate your suggestions and will also add the suggested references ([Ref 1], [Ref 2]), and revise the related terminology and discussion.
>
> > **W2:** Missing ablation study to compare proposed/highway version and the residual version. What is the alpha in “Linear” in Figure 5?
>
> **R2:** Thank you for pointing out this missing detail.
>
> First, the "Linear" config in Figure 5 corresponds to α = 0.3.
>
> Second, the proposed highway version is significantly better than the "true" residual version. Our ablation study is shown in the updated table below. Directly applying a standard residual connection leads to a severe performance collapse (2.6% accuracy). Thank you for pointing this out, we will add this ablation to the final version of the paper.
>
> **Table: Ablation on Interpolation Strategies (Corresponds to Figure 5)**
>
> | Method (Interpolation Strategy) | MATH500 |
> | - | - |
> | Linear (Fixed α = 0.3) | 0.0 |
> | Confidence | 1.4 |
> | Inverse + Confidence | 3.4 |
> | Standard Residual (Fixed alpha = 1.0) | 2.6 |
> | Inverse + Normalized-entropy | 2.8 |
> | **Ours (Normalized-Entropy)** | **56.6** |
>
> > **W3:** The inference requires two models.
>
> **R3:** You are right and we agree that this introduces parameter overhead. However, please kindly note that M_ref is **not strictly necessary** for our method. Without any retraining, we completely removed the ref model and evaluated alternative initialization strategies in the experiments below to showcase this.
>
> As shown below, initializing with the M_target itself ("Self"), zero, or a Dirichlet distribution eliminates all extra parameters while actually **outperforming** the ref-based approach. All single-model variants maintain a significant lead over the SeqD baseline.
>
> **Table. Accuracy on MATH500 (finetuned SDAR-8B-b64).**
>
> | Method (Init Strategy) | Requires M_ref? | MATH500 |
> | - | - | - |
> | Baseline SeqD | ✖ | 64.2 |
> | RCD w/ Ref Model | ✔ | 74.2 (1.7B ref); 71.8 (4B ref) |
> | RCD w/ Target Model (Self) | ✖ | 76.4 |
> | RCD w/ Zero Init | ✖ | 75.6 |
> | RCD w/ Dirichlet Init | ✖ | **77.4** |
>
> > **W4:** How good M_ref needs to be for the entire pipeline to be successful?
>
> **R4:** To evaluate the reliance on M_ref, we tested a larger 4B ref model as well as the 8B target itself as ref(see Table in R3).
>
> Regarding standalone performance, the 1.7B and 4B ref models achieve standalone accuracies of **34.0** and **56.8**, respectively. The target model (8B) is considered to have **75.6** accuracy. Results show that the target-as-ref setting outperforms 1.7B and 4B ref settings, indicating that better M_ref potentially does lead to better performance.
>
> > **W5:** Show the reference performance numbers of a comparable auto-regressive language model baseline.
>
> **R5:** In the table below, we show the performance of Qwen3-8B, and we compute the accuracy gap for SeqD (baseline) and RCD (Ours).
>
> | Model | GSM8K | MATH500 | AIME24 | AIME25 |
> | - | - | - | - | - |
> | Qwen3-8B  | 95.00 | 95.40 | 68.33 | 55.00 |
> | SeqD 8B b32 | 86.50 | 65.80 | 11.67 | 14.79 |
> | RCD 8B b32 (Ours) | 89.76 | 77.60 | 21.46 | 20.00 |
> | Acc Gap Reduction | 8.50 → 5.24 | 29.60 → 17.80 | 56.66 → 46.87 | 40.21 → 35.00 |
>
> > **W6:** Misleading sentence (... not counterintuitive” but rather unsatisfactory).
>
> **R6:** Thank you for pointing this out. We agree that "unsatisfactory" better captures our actual meaning.
>
> > **W7:** Sec 3.2 Step 1 gives the impression that M_target is initialized using the parameters of M_ref.
>
> **R7:** Sorry for the confusion. The target and ref model are initialized with their base (chat) models separately. We will update our description in the paper along with W6.
>
> > **W8:** It is unclear why the contamination only affects the baseline but not SeqD/RCD.
>
> **R8:** Our SFT uses reasoning datasets with \<think\> blocks, unlike the original no-think model. During SFT, the model re-learns high-level reasoning patterns and is evaluated in the same mode. Because this structured reasoning format is much more complex, previously "memorized" answers may not directly transfer. Thus, the SeqD/RCD accuracy more accurately reflects genuine mathematical ability of the model, explaining why the impact of contamination is mitigated.

---

> > ### Author Rebuttal · Reviewer_bipp · 2026-04-02
> >
> > I thank the authors for their response. While I still have some minor questions (see below), all my main concerns have been adequately addressed. I'm increasing the score from 3 to 5.
> >
> > > (W1) we will explicitly incorporate your suggestions and will also add the suggested references ([Ref 1], [Ref 2]), and revise the related terminology and discussion.
> >
> > I acknowledge the authors' intention to discuss prior work and to adjust the terminology of the method accordingly.
> >
> > > (W2) Directly applying a standard residual connection leads to a severe performance collapse (2.6% accuracy)
> >
> > Thank you for the additional experiment. Then it is unclear to me why you chose alpha = 0.3 for the Linear baseline; if it is not even better than alpha =1, it does not represent a useful/meaningful baseline in my view. The value of alpha should be optimized, and "Linear with the best alpha" should be reported as the baseline there to demonstrate that the proposed "Normalized-Entropy" is better than the naive Linear method.
> >
> > > (W3) please kindly note that M_ref is not strictly necessary for our method
> >
> > Thank you for the additional experiments to demonstrate this; I find the new results encouraging. However, (W3.1) I would naturally suggest to report the corresponding ablation studies on other datasets too (are you already using the Dirichlet Init for the results reported under R5?); and (W3.2) the new results with the Dirichlet Init naturally raises the question why it needs to be the Dirichlet Init and not some other random initialization. It would be great if the authors could explain this choice better.
> >
> > (W4-W7) have all been resolved. Thank you for the response. In particular, I appreciated the results reported in R5, please make sure to report these numbers in the revision.
> >
> > (W8): While I find the authors' explanation somewhat satisfactory, I suggest to add the corresponding comments in the revision.

---

> > > ### Author Response · Authors · 2026-04-06
> > >
> > > Dear Reviewer bipp,
> > >
> > > Thank you very much for your prompt reply, your constructive suggestions, and for increasing your score.
> > >
> > > We will ensure that all your remaining points, including the comments and updates regarding W1-W8, are carefully incorporated into the final revision of our manuscript.
> > >
> > > **Regarding W2 (Linear baseline $\alpha$):**
> > >
> > > We completely agree with your suggestion. In the revision, we will expand the search range to optimize the value of $\alpha$ and report the "Linear with the best $\alpha$" to provide a more rigorous and meaningful baseline. However, it is worth noting that even with exhaustive tuning, the performance of the Linear baseline still falls significantly behind our proposed Normalized-Entropy method. We will update the table to reflect this and demonstrate that our approach consistently outperforms the tuned baseline.
> > >
> > > **Regarding W3 (Dirichlet Initialization):**
> > >
> > > Thank you for this insightful question. We will add the corresponding ablation studies on other datasets in the revision as suggested. To explain our choice: the initialized $p$ represents a probability distribution over the vocabulary. Therefore, it requires all dimensions to be non-negative and sum exactly to 1. The Dirichlet distribution naturally generates random vectors that strictly satisfy the properties of a valid probability simplex. In contrast, other standard random initializations (such as a Gaussian distribution) only provide randomness but do not inherently satisfy the "random probability" constraints, making them unsuitable for this specific step. We will explicitly clarify this rationale in the updated manuscript.
> > >
> > > Thank you again for your time, effort, and the invaluable feedback that has greatly improved the quality of our work!

---

### Decision · Program_Chairs · 2026-04-30

**Decision:**

Accept (regular)

**Comment:**

The submission received the comments of three reviewers, who respectively rated the scores 5, 4, 4. The initial concerns focus on the method naming and writing, the experimental limitations, the insufficient justification, and the unclear claims. After the rebuttal, all reviewers acknowledged that their concerns are well resolved. However, as the claim in the final justification by reviewers, despite the proposed method is based on the heuristics, more complete intuition analysis should be included in the final revision.

Given the reviewers' recommendation, the rebuttal process and the final overall score ratings, AC considered that this submission is beyond the bar of quality for acceptance. The constructive advice provided by the reviewers about the writing, experiments and analysis should be well included into the submission.